# ROR2 Is Epigenetically Regulated in Endometrial Cancer

**DOI:** 10.3390/cancers13030383

**Published:** 2021-01-21

**Authors:** Dongli Liu, Luis Enriquez, Caroline E. Ford

**Affiliations:** Gynaecological Cancer Research Group, Lowy Cancer Research Centre, School of Women’s and Children’s Health, Faculty of Medicine, University of New South Wales, Sydney, NSW 2052, Australia; dongli.liu@unsw.edu.au (D.L.); LEnriquez@redcrossblood.org.au (L.E.)

**Keywords:** ROR2, endometrial cancer, methylation

## Abstract

**Simple Summary:**

Endometrial cancer is one of the fastest rising cancers in women. The Wnt signalling receptor ROR2 has been shown to play distinct roles in regards to tumorigenesis in different tumour types. The aim of this study was to investigate the role of ROR2 in endometrial cancer and to determine if *ROR2* expression is epigenetically regulated. Through the analyses of publicly available TCGA and GEO datasets, low *ROR2* expression was correlated with unfavourable outcome and reduced overall survival of endometrial cancer patients. In addition, we observed epigenetic repression of *ROR2* expression in endometrial cancer cell lines and patient samples. Ectopic expression of ROR2 in vitro inhibited the invasive ability of high grade serous endometrial cancer cells. Therefore, we concluded that ROR2 plays a tumour suppressor role in endometrial cancer and appears to be a diagnostic or therapeutic candidate.

**Abstract:**

The Wnt signalling receptor ROR2 has been identified as a possible therapeutic target in numerous cancers; however, its exact role remains unclear. The aim of this study was to investigate the role of ROR2 in endometrial cancer (EC) and the potential mechanism associated with its altered expression. The association between ROR2 mRNA expression levels and clinicopathological parameters, including overall survival (OS), in EC was analysed in The Cancer Genome Atlas Uterine Corpus Endometrial Carcinoma (TCGA-UCEC) cohort and GEO dataset GSE17025. Four EC cell lines (KLE, MFE-296, Ishikawa and ARK-1) and eight clinical EC samples were analysed for *ROR2* methylation via Combined Bisulphite Restriction Analysis (COBRA) and bisulphite genomic sequencing (BGS). In addition, the functional effects of ROR2 overexpression were investigated in Ishikawa and ARK-1 cells following ectopic ROR2 expression. *ROR2* promoter methylation or reduced *ROR2* expression were both found to correlate with shorter OS, high grade and serous subtype in the TCGA-UCEC and GEO datasets. *ROR2* was epigenetically silenced by promoter methylation in both patient samples and cell lines. A significant correlation between *ROR2* expression levels and promoter methylation was observed in patient samples (r = −0.797, *p* = 0.018). ROR2 restoration in ARK-1 significantly decreased invasion ability, with associated changes in epithelial-mesenchymal transition (EMT) markers. ROR2 plays a tumour-suppressor role in EC and is epigenetically suppressed with the development of disease. It may represent a diagnostic or therapeutic candidate for EC.

## 1. Introduction

Endometrial cancer (EC) is one of the fastest rising cancers worldwide, with more than 380,000 new cases diagnosed in 2018 [1]. It was historically classified into two subgroups: estrogen-dependent Type I (mainly endometrioid subtype) and estrogen independent Type II (other subtypes such as serous, clear cell) [2]. Recent multi-omics characterisation of EC led by The Cancer Genome Atlas (TCGA) identified four molecular subtypes: POLE-mutant tumours (ultrahypermutated), microsatellite instability hypermutated, copy-number low and copy-number high tumours [3]. The copy-number high cluster included most of the serous and serous-like endometrioid subtypes while the other three clusters mainly consisted of the endometrioid subtypes. Although the TCGA classification holds great potential, this genetic information has largely not yet been translated into changes in clinical practice. The conventional treatment for EC is total hysterectomy and bilateral salpingo-oophorectomy [4], combined with adjuvant radiotherapy for high-risk and aggressive cases [5], albeit the recent PORTEC-3 clinical trial showed that only p53 mutant EC patients could benefit from the adjuvant therapy [6]. The effectiveness of surgery along with frequent early detection for early stage low grade endometrioid EC contributed to a survival rate over 80% in the period 2008 to 2014 [7]. However, survival outcomes for high grade, metastatic endometrioid EC as well as highly aggressive subtypes, remain poor [8]. This is largely due to the fact that most clinically aggressive subtypes are diagnosed at late stages, and respond poorly to the available treatment options. The rapidly rising incidence of aggressive non-endometrioid EC [9] has necessitated the identification of new biomarkers to predict EC progression, and target therapeutically.

The Wnt signalling pathway (encompassing β-catenin dependent and independent arms) has been implicated in a range of cancers. The canonical Wnt/β-catenin pathway plays an essential role in the hormonally regulated menstrual cycle and aberrant activation of this pathway has been associated with the tumorigenesis and progression of EC [10,11,12]. Mutations in β-catenin as well as nuclear localisation (indicative of a hyperactivated pathway) have been observed extensively in endometrioid but not in non-endometrioid EC [13,14]. However, mechanisms associated with the abnormal nuclear accumulation of β-catenin in subtypes of EC lacking β-catenin mutations [15] remain unclear.

Compared to the β-catenin-dependent (or “canonical”) pathway, studies investigating the independent pathway in EC remain limited. This pathway can be activated through Wnt5a ligand binding to the tyrosine kinase-like orphan receptors ROR1 and ROR2. Different to the oncogenic role of the two receptors in ovarian cancer where both receptors are overexpressed and associated with survival [16], our previous study in EC suggested distinct roles for ROR1 and ROR2, with high ROR1 and low ROR2 expression associated with shorter survival [17]. ROR2 is involved in both arms of the Wnt-signalling pathway. It activates β-catenin independent signalling upon binding with Wnt5a [18,19] and inhibits Wnt/β-catenin dependent signalling via interacting with Wnt5a [20]. It can also trigger the canonical arm by binding with Wnt3a [21,22]. It has been reported to play distinct roles in regards to tumorigenesis in different tumour types [23]. In gynaecological cancers, ROR2 has been shown to be upregulated in ovarian cancer patients compared to benign cases [16], and high ROR2 expression was significantly correlated with poor prognosis in cervical cancer [24]. In contrast, our previous study suggested a potential tumour suppressor role for ROR2 in EC [17].

Epigenetic inactivation of the *ROR2* promoter has been observed in colorectal cancer where ROR2 acts as a tumour suppressor [25,26]. Hypermethylated *ROR2* regulated epithelial-mesenchymal transition (EMT) and downstream cell migration and invasion through promoting the Wnt/β-catenin pathway [27]. There have also been concerns raised about the quality and specificity of antibodies used to detect ROR2 in previous studies [28], and potential differences between the importance of expression at the mRNA rather than the protein level. Moreover, as a pseudokinase, how ROR2 regulates signalling remains unclear. A recent study suggested that it might undergo conformational transitions to interact with binding ligands, similar to that of insulin receptor kinase (IRK) [29]. Therefore, it can be seen that more research is required to elucidate the function of ROR2 in EC.

In this study, the role of ROR2 and its regulation in EC was investigated through bioinformatics, epigenetic analysis of patient samples and functional analysis of *ROR2* expression in vitro.

## 2. Results

### 2.1. Expression and Methylation Status of ROR2 is Associated with Overall Survival in an EC Cohort

We analysed expression of *ROR2* in The Cancer Genome Atlas Uterine Corpus Endometrial Carcinoma (TCGA-UCEC) cohort and found that patients with high *ROR2* expression or low methylation at three specific CpG sites located on the *ROR2* promoter showed significantly higher overall survival (OS) compared to those with low *ROR2* expression or high methylation of CpG sites (Figure 1A–D). For the multivariable analysis, stage and methylation at the two of the three CpG sites (cg01062029 and cg14145355) were significantly correlated with OS (Figure 1E).

Compared to adjacent normal tissue samples (*n* = 52), the gene expression level of *ROR2* was significantly lower in tumour tissue (*n* = 553) in the TCGA-UCEC cohort (Appendix A). However, no significant difference was observed between tumour and normal tissue in the methylation level of any of the three CG sites within the *ROR2* promoter (Appendix A).

### 2.2. ROR2 Was Epigenetically Suppressed in High-Grade EC in Public Datasets

Significantly negative correlations between ROR2 mRNA expression level and two of the three CG sites within the *ROR2* promoter were observed in the TCGA-UCEC cohort (Figure 2A). We next explored whether *ROR2* expression or methylation was associated with stage or grade. The TCGA-UCEC cohort contained samples across all stages of EC, while the GSE17025 only included stage I samples so associations with stage were unable to be performed. While a trend was seen towards lower expression in higher stages, no significant difference was observed for *ROR2* expression between each of the two adjacent stages (Figure 2B). However, *ROR2* expression was significantly decreased in high grade compared to low grade EC in both the TCGA-UCEC and GSE17025 cohorts (Figure 2C,E). Methylation level of the three CpG sites in the *ROR2* promoter region was significantly higher in the high grade compared to low grade (Figure 2D). In the high-grade subgroup of the TCGA-UCEC cohort, methylation level of two CpG sites cg14145355 and cg03900646, rather than *ROR2* expression level, were significantly associated with overall survival after filtering against other variables including age, BMI, stage (Appendix A, *p* = 0.013 and 0.014 respectively). *ROR2* expression also showed a negative correlation with methylation level of 2 of the three CpG sites located on the *ROR2* promoter in this subgroup (Appendix A).

### 2.3. ROR2 Was Epigenetically Suppressed in Serous EC in Public Datasets

We next investigated whether *ROR2* expression or methylation was associated with the two main subtypes of EC: serous and endometrioid. Histological subtype data were available for both the TCGA-UCEC and GSE17025 cohorts. The serous subtype showed significantly lower *ROR2* expression levels (Figure 3A,B) and a higher methylation level of two out of three CG sites (Figure 3C) compared to the endometrioid subtype in both cohorts.

In multivariable analysis, one CpG site cg01062029 located on the *ROR2* promoter was significantly associated with overall survival in serous EC patients of the cohort (Appendix A, *p* = 0.033). ROR2 expression showed a negative correlation with methylation level of the CpG site cg01062029 (Appendix A).

In addition, when the cohorts were split into the two main histologic subtypes of EC (serous or endometrioid), the downregulation of *ROR2* in high grade compared to low grade was observed in the subgroup of endometrioid but not serous EC (Appendix A).

### 2.4. Expression of ROR2 was Regulated by Promoter Methylation in EC Cell Lines and Patient Samples

In order to determine if *ROR2* expression was epigenetically regulated by promoter methylation in EC, we initially designed a COBRA assay to measure methylation at 3 key CpGs in the *ROR2* promoter. To further quantify the extent of methylation across the region analysed in the COBRA assay, bisulphite genomic sequencing (BGS) was also performed. Four EC cell lines and 8 patient samples were analysed in this study. The EC cell lines ARK-1 and KLE have low expression of *ROR2*, and showed digestion in the COBRA gel compared to the EC cell line MFE296, which has high *ROR2* expression (Figure 4A). Consistently shown in the BGS result (Figure 4B), MFE296 showed an extremely low methylation level of the CpG sites in the region. The poorly differentiated cell line KLE and high-grade serous cell line ARK-1 both showed a much higher methylation level compared to the other two low grade endometrioid cell lines. Both qRTPCR and Western blot consistently showed undetectable *ROR2* expression in KLE and ARK-1 (Figure 4C,D). Ishikawa showed a moderate level of *ROR2* expression and a medium level of methylation from the BGS result.

The same analysis was conducted on a small cohort of 8 EC patient samples (7 endometrioid and 1 serous). The serous EC patient (HSA2405) showed an obviously distinct digestion pattern compared to others with endometrioid subtypes (Figure 5A), and highest methylation level of the *ROR2* promoter region (Figure 5B). This sample also had undetectable *ROR2* expression compared to other endometrioid samples, as measured by qRTPCR (Figure 5C). A significant correlation between the expression level of *ROR2* and methylation was observed (*r* = −0.797, *p* = 0.018) from Pearson’s correlation test. These results confirmed *ROR2* was suppressed in high grade serous EC patients as well as cell line, which was correlated with a promoter methylation (Figure 4 and Figure 5).

### 2.5. ROR2 Overexpression Inhibited Cell Invasion in ARK-1

As the bioinformatics and clinical cohort analysis highlighted the potential importance of ROR2 in higher grade and serous EC, we next performed in vitro functional analysis of ROR2 re-expression in two EC cell lines. The ARK-1 high-grade serous (Type II) EC cell line was selected for subsequent experiments due to its low *ROR2* expression and high *ROR2* promoter methylation index. The Ishikawa endometrioid (Type I) EC cell line was selected for its moderate ROR2 expression and moderate ROR2 promoter methylation index. ROR2 plasmid transfection was shown to be effective after 48 h in both Ishikawa and ARK-1 via qRTPCR and Western blot results (Figure 6A). Compared to the empty vector control, ROR2 overexpression reduced proliferation (Figure 6B) and migration (Figure 6C, *p* = 0.140) of ARK-1, but this inhibition did not reach statistical significance. However, ROR2 overexpression in ARK-1 did significantly inhibit invasion (Figure 6D, *p* = 0.020). No significant effect was observed in the Type I EC cell line Ishikawa in terms of proliferation, migration or invasion ability. Increased E-cadherin and decreased Vimentin levels were observed after ROR2 overexpression in ARK-1 (Figure 6E), suggestive of a potential effect on EMT.

## 3. Discussion

As the fifth most common cause of cancer in women worldwide, there has been surprisingly little research conducted into the key genes associated with EC. We have previously identified the Wnt receptor, ROR2, as a gene of potential interest in EC [17], and have also shown that it can be epigenetically regulated in the development of colorectal cancer [26].

In this study, we combined a bioinformatic analysis of publicly available datasets, with in vitro analysis, utilising both verified EC cell lines and primary patient samples. This allowed us to combine the statistical power of analysing large cohorts of samples with a smaller defined patient cohort where we could control parameters to analyse expression and methylation. On top of this we investigated the effect of manipulation of ROR2 expression levels on the metastatic potential of two EC cell lines.

We found that *ROR2* was downregulated in EC tumour tissue compared to the adjacent normal tissue, and that this gene silencing was associated with tumour grade. Low expression or hypermethylation of *ROR2* was correlated with significantly lower overall survival in EC patients. In addition, the methylation of 2 out of 3 CpG sites within the *ROR2* promoter was associated with OS after stratifying by other parameters including stage, grade and histology subtypes.

Consistent in both datasets (TCGA-UCEC and GSE17025 cohorts), ROR2 was epigenetically suppressed in high grade and serous EC compared to low grade and endometrioid EC. This suggests that the epigenetic regulation of ROR2 may develop along with the growth of EC. This is of relevance to our earlier analysis of colorectal adenomas and colorectal cancer where we observed a stepwise reduction in ROR2 expression from normal colon tissue, through pre-malignant ademonas through to cancer [26,28]. We found that no significant difference in ROR2 expression was observed between low and high grade serous EC in either dataset, however the biased distribution of grades in serous EC samples (4 low, 126 high in TCGA-UCEC and 8 low, 9 high in GSE17025) suggests that statistical analysis was not valid. Future analysis is warranted with the involvement of more low-grade (grade 1 or 2) serous EC samples, though due to their clinical rarity this will be challenging.

The four EC cell lines selected for this study incorporated both Type I and II and ranged from grade 1 to grade 3. The Ishikawa cell line was derived from a grade 1, Type I endometrial adenocarcinoma [30], although this tends to transform into undifferentiated morphology after long-term culture [31]. MFE296 was established from a grade 2, Type I endometrial adenocarcinoma [32]. KLE is classified as a G3 (poorly differentiated) endometrial carcinoma [33], but whether KLE belongs to Type I or II remains controversial [34,35,36]. ARK-1 is a high grade Type II endometrial serous carcinoma [37]. As was have seen in the public EC patient cohorts, ROR2 was barely detectable in high grade KLE and serous ARK-1 at both transcriptional and translational levels. Although different regions were targeted in the *ROR2* promoter between the HumanMethylation450 and bisulphite genomic sequencing, hypermethylation was also observed in high grade and serous EC cell lines. Generally, the three probes from HumanMethylation450 and the region analysed by bisulphite sequencing covered an 857bp region (653 downstream and 204 upstream the TSS of ROR2), which in combination served as a comprehensive representative for the CpG rich promoter region of ROR2.

In our own patient cohort, the serous EC sample showed an extraordinarily low expression level of *ROR2* compared to the other endometrioid samples, in line with its highest methylation level in the ROR2 promoter region among the 8 patients. In combination with the large public cohorts and well-established EC cell line models, this cohort suggested deregulation of ROR2 expression as well as hypermethylation of ROR2 in EC. As this may have implications for the serous and high grade subtypes of EC, of which we had a limited number in this local cohort, we therefore recommend future investigation of a larger cohort of primary patient samples including more serous EC samples to uncover the epigenetic role in progression of EC. In addition, our clinical cohort incorporated one sample of Malignant Mixed Müllerian Tumour (MMMT). The majority of MMMT cases share common molecular features with serous EC while a lesser proportion was similar to endometrioid subtype [38]. The DNA methylation profiles of MMMT also grouped the subtype into three clusters, with one similar to endometrioid EC and the other two resembles serous EC [39]. In our clinical cohort, the methylation and expression of ROR2 of the MMMT sample is similar to that of the endometrioid EC samples. However, due to the heterogeneity of this subtype, it would be helpful to include more MMMT subtypes in future studies to uncover epigenetic regulation mechanisms.

Following ROR2 restoration via ectopic expression, the type II EC cell line ARK-1 decreased aggressive features, especially invasion ability. The inhibition of the epithelial–mesenchymal transition (EMT) shown as an increased E-cadherin or reduced Vimentin level appears to be the main associated mechanism. Compared to ARK-1, Ishikawa demonstrated lower proliferation, migration or invasion ability. This could be due to suppression in the EMT process, which could be supported by the high E-cadherin or low Vimentin expressed in Ishikawa compared to ARK-1. As EMT has been linked with tumour invasion and metastasis [40,41], it could also explain the different metastatic spread pattern of Type I and II EC.

ROR2 has been reported to act as a tumour promoter or suppressor depending on tumour type [23]. The bidirectional function of ROR2 could be derived from the dual regulations in two arms of the Wnt signalling pathway. In osteosarcoma where ROR2 acts as an oncogene, downregulation of ROR2 inhibited cell invasiveness through the non-canonical (β-catenin independent) Wnt signalling pathway [42]. The presumed ligand for ROR2 in endometrial cancer is Wnt5a. However, a previous study in colon cancer showed that restoration of the tumour suppressor ROR2 impaired tumour growth through inhibition of β-catenin-dependent Wnt signalling, in a Wnt5a independent manner [25]. Therefore, ROR2 may dominate in either of the two arms of downstream Wnt signalling depending on the tumour context. Further exploration into the role of ROR2 in either arm of the Wnt signalling as well as its ligands is required in EC. As ROR2 shows a similar role in EC as in colon cancer, the epigenetic repression of ROR2 could be a possible explanation for the abnormal nuclear accumulation of β-catenin in subtypes of EC lacking β-catenin mutations.

A recent study demonstrated the potential application of DNA methylation profiles of certain biomarkers for EC screening of cervical scrapings [43]. However, the biomarkers identified in the study were based on Type I EC, which left the application in high grade or other Type II subtypes unclear. Our study proposes a promising biomarker for methylomic analysis in high grade and serous EC, which warrants future exploration.

Genes harbouring promoter hypermethylation could be restored through a demethylation agent such as Azacitidine. In addition, demethylating agents, either alone or in combination with other drugs, have been applied in clinical trials and have shown promising outcomes [44,45,46,47]. Our previous in vitro study in EC showed ROR2 overexpression in combination with silencing its sister receptor ROR1 inhibited the metastatic potential of KLE endometrial cancer cells more than either of the modifications alone [48]. With several ROR1-targeting therapies currently in development and in phase I clinical trials for other cancers, combination therapy may hold great potential in patients with HGSEC, who currently have limited therapeutic options. However, the functional outcome associated with the global demethylation effect remains unclear. Despite the global hypomethylation of EC compared to normal endometrium, promoter hypermethylation of several genes such as MLH1 (DNA mismatch repair gene) [49], PTEN (Phosphatase and ten-sin homolog) [50] and APC (which regulates β-catenin Wnt signalling pathway) have been observed frequently in certain types of EC, especially Type I. Compared to Type I, Type II EC showed distinct DNA methylation profiles, including a different methylation pattern of DNMT (DNA methyltransferase gene) [51]. Therefore, divergent responses to the demethylation agents between Type I and II EC patients could be expected. Previous in vivo studies have demonstrated promising effects of the demethylating agent 2′-deoxy-5-azacytidine (DAC) in restoring MLH1 expression and reversing resistance to chemotherapies in ovarian cancer [52]. There may be potential for combination treatment with chemotherapy and demethylation agents to treat chemo-resistant EC patients, but further research into this area will be required.

To sum up, ROR2 plays a tumour suppressor role in EC and is epigenetically suppressed with development of the disease, which potentially serves as a predictive biomarker for prognosis or a therapeutic target in EC patients.

## 4. Materials and Methods

### 4.1. TCGA-UCEC Cohort

The TCGA-UCEC cohort was analysed for this study. The gene expression and methylation level of *ROR2* as well as clinicopathological data including the International Federation of Gynecology and Obstetrics (FIGO) stage, histological subtype, tumour grade, survival status and time were extracted from the UCSC Xena platform [53] on 8th of April, 2020.

The clinicopathological parameters of the cohort are summarised in Appendix A. The cohort incorporates 553 EC cases, with 425 cases available for gene expression data and among which, there are 395 cases for methylation information. The gene expression level of *ROR2* was measured using HiSeq 2000 RNA sequencing platforms as well as the DNA methylation profile of *ROR2* promoter using the Illumina Infinium HumanMethylation450 platform. The methylation level (Beta value) of three probes within the *ROR2* promoter region—cg01062029 (TSS1500), cg03900646 (TSS200) and cg14145355 (TSS1500)—were extracted for the analysis. Tumour grade was aggregated to low (G1 and G2) and high (G3 and high grade in general).

### 4.2. GEO Dataset

The clinicopathological parameters of the cohort are summarised in Appendix A. The GEO dataset GSE17025 [54] with heterogeneous distribution of grade and histological subtypes of EC cases (*n* = 91), was analysed. The GEOquery package [55] of R was used to retrieve the expression matrices and clinicopathological parameters of the datasets, the expression data of GSE17025 was normalised to a target intensity of 500 using Affymetrix’s MAS5.0 and was log2 transformed. Tumour grade 1 and 2 were grouped as low, grade 3 as high in the analysis.

### 4.3. Cell Culture

EC cell lines ARK-1, Ishikawa, MFE296 and KLE were selected for distinct ectopic expression level of ROR2. ARK-1 was kindly provided by Dr Alessandro Santin (Yale University, New Haven, CT, USA). Ishikawa was a gift from Associate Professor Jeff Holst (UNSW, Sydney, Australia). KLE and MFE296 were a gift from Associate Professor Kyle Hoehn (UNSW, Sydney, Australia). Cell lines were maintained in medium containing 10% foetal bovine serum (FBS), 1% GlutaMAX and 1% penicillin/streptomycin and kept in 5% CO_2_ at 37 °C. Specifically, KLE was cultured in DMEM/F12 media, ARK-1 was cultured in DMEM media. Ishikawa and MFE296 were cultured in MEM media. Cell lines underwent mycoplasma testing once a month and were validated at the cell line identification service at the Garvan Institute of Medical Research (Sydney, Australia).

### 4.4. Patient Samples

High quality DNA and RNA samples from eight EC patients (defined as 260/280 absorbance ratio of 1.7–2.0 and 260/320 absorbance ratio >1.5, RNA Integrity Number >8 respectively) were acquired from the Health Science Alliance (HSA) Biobank, UNSW. The clinicopathological parameters for the 8 patients are listed in Table 1.

### 4.5. DNA and RNA Extraction

DNA and RNA of the EC cell lines were extracted simultaneously using the All-In-One DNA/RNA Miniprep kit (Astral scientific, Taren Point, Australia).

### 4.6. Combined Bisulphite Restriction Assay (COBRA)

DNA samples extracted from both the cell lines and patient tumour samples were bisulphite converted using the EpiTect Bisulfite Kit (#59104, Qiagen, Hilden, Germany) as per the manufacturer’s instructions. A 395 bp region including 3 TCGA/CCGA restriction sites within the *ROR2* promoter region was amplified by the primers (F-AGGAAATGTTTAGGAAAATAAATAGGT, R-AAAACAAACAACTAAAATACTAAAAA) and digested with TaqI restriction enzyme mix (#R0149S, NEB, Ipswich, MA, USA). The targeted region as well as the two CpG sites (cg03900646 and cg14145355) were mapped on the region 0.5 kb upstream and 1.0 kb downstream of *ROR2* transcriptional start site (TSS) shown in Appendix A. Digested samples and corresponding non-digested controls were loaded on to a 1.5% agarose gel supplemented with GelRed (#41003, Biotium, Fremont, CA, USA), subjected to electrophoresis and visualised under UV light.

### 4.7. Bisulphite Sequencing

The same PCR product from the COBRA assay was ligated into pCR2.1 vector and transformed in One Shot competent cells using the TA Cloning kit (Life Technologies, Carlsbad, CA, USA) as per the manufacturer’s instructions. For each colony PCR, up to 10 colonies were picked for plasmid extraction using GeneJET plasmid miniprep kit (Thermofisher Scientific, Waltham, MA, USA). Plasmids with ROR2 PCR inserts were sent for Sanger sequencing at the Ramaciotti Centre, UNSW using the BigDye system with M13 Reverse and Forward primers.

### 4.8. qRTPCR

Real-time reverse transcriptase PCR was performed on the RNA sample of 8 patients as previously described [16]. The relative expression level of *ROR2* was calculated using 2^–∆∆Ct^ method and normalised against the mean of three house-keeping genes (*HSPCB, SDHA, RPL13A*). Primers were provided in [16].

### 4.9. Western Blot

Total protein lysates of the cell lines were made with cell lysis buffer (Cell Signalling Technology, Danvers, MA, USA) containing a protease inhibitor (Sigma-Aldrich, St. Louis, MO, USA). Up to 20 μg of the protein samples were loaded for Western blotting as previously described [16]. The primary antibodies used were anti-ROR2 (#34045, QED Bioscience, San Diego, CA, USA), anti-α-Tubulin (#3873, Cell Signalling, Danvers, MA, USA), anti-Vimentin (#5741s, Cell Signalling, Danvers, MA, USA) and anti-Ecadherin (#3195s, Cell Signalling, Danvers, MA, USA). The normalized band intensities, as well as the uncropped blots, were provided in Appendix A.

### 4.10. ROR2 Transfection

Ishikawa and ARK-1 were selected for ROR2 transfection experiments as a model for Type I and II EC, respectively. Cells were transfected with either 500ng ROR2 pFLAG plasmid (constructed as described previously [26]) or empty pFLAG-CMV-4 plasmid using Lipofectamine 2000 (Life Technologies, Carlsbad, CA, USA). After 6 h, transfection mix was removed and replaced with complete media.

### 4.11. Proliferation Assay

Proliferation was determined at 24 h, 48 h and 72 h after transfection using the Cell Counting Kit-8 (CCK-8, Sigma-Aldrich, St. Louis, MO, USA) as per the manufacturer’s protocol. For ARK-1, 2000 cells were plated while 4000 cells were plated for Ishikawa per well in the 96-well plate six hours after transfection.

### 4.12. Migration and Invasion Assay

Corning transwell inserts and Matrigel pre-coated transwell inserts (Corning Life Sciences, Tewksbury, MA, USA) were used for migration and invasion analysis respectively as per the manufacturer’s protocol. For Ishikawa, 5 × 104 cells were plated in the upper chamber of each insert and incubated for 48 h for both migration and invasion assays. For ARK-1, 2 × 10^4^ or 5 × 10^4^ cells were plated for migration or invasion for 24 h following transfection.

### 4.13. Statistical Analysis

Paired *t*-test was performed to estimate the significance between matched normal and tumour tissue from the TCGA-UCEC cohort. Statistical significance of expression and methylation level of *ROR2* between tumour grades, histological subtypes and stages was carried out using unpaired *t*-test. Correlation between expression and methylation was analysed using Pearson’s coefficient (R). Kaplan–Meier curves were produced for overall survival (OS) analyses. The optimal cut-point for variables was applied with the maximally selected rank statistics from the “maxstat” package in R. The log-rank test was used to evaluate the association between the covariates and OS. Cox multivariate regression including age, BMI, FIGO stage, grade and subtypes was also applied on the OS. Data were presented as mean ± standard deviation (SD). For the in vitro assays, all experiments were repeated three times. Results were shown as mean ± standard deviation, unpaired *t*-test was used to compare the two conditions. All the analyses were performed using R (v3.6.3). Figures were provided in R (v3.6.3) and GraphPad Prism (v7.02). Significance was defined at *p* < 0.05.

## 5. Conclusions

This study is the first to investigate the role and epigenetic regulation of ROR2 in EC. ROR2 plays a tumour-suppressor role in EC and is epigenetically suppressed with the development of disease. It may therefore be a diagnostic or therapeutic candidate for EC.

## Figures and Tables

**Figure 1 cancers-13-00383-f001:**
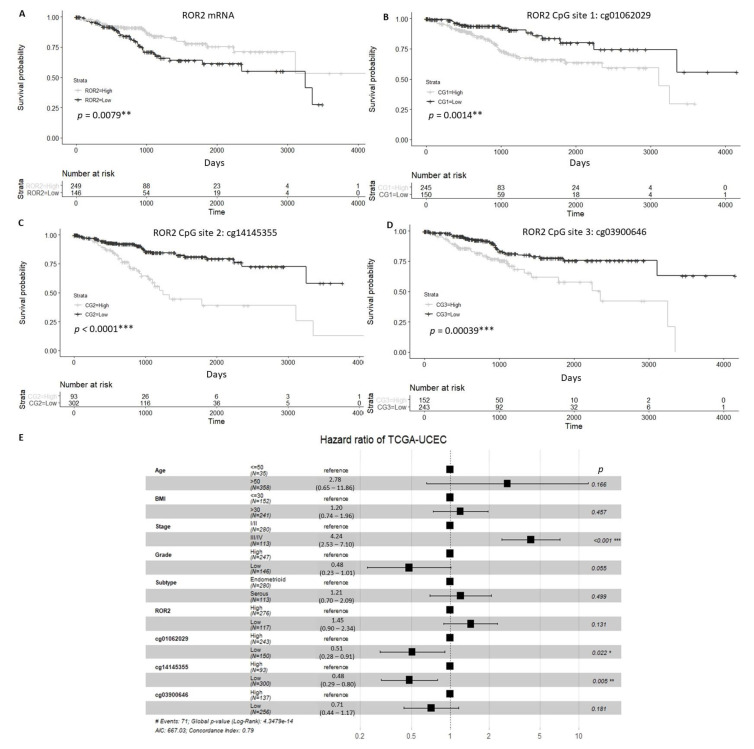
Univariate and multivariable overall survival analysis of TCGA-UCEC cohort patients. (**A**). Patients with low ROR2 expression showed significantly worse overall survival (OS) compared to those with high ROR2 expression. (**B**–**D**). High methylation levels of three CpG sites (CG1-cg01062029, CG2-cg14145355, CG3-cg03900646) located on the ROR2 promoter were associated with worse OS significantly (*p* = 0.001, *p* < 0.001, *p* < 0.001 respectively) (**E**). Forest plot of Cox regression result incorporated age, BMI, stage, grade, subtype, expression and methylation of *ROR2* on OS. * Significant at *p* < 0.05 level. ** Significant at *p* < 0.01 level. *** Significant at *p* < 0.001 level.

**Figure 2 cancers-13-00383-f002:**
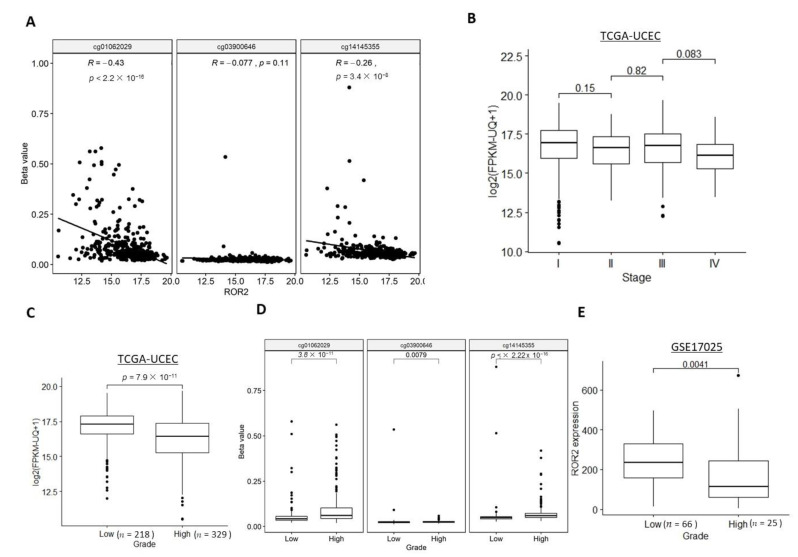
ROR2 mRNA expression level was downregulated in high grade EC due to hypermethylation of ROR2 promoter region in the TCGA-UCEC and GSE17025 datasets. (**A**). *ROR2* expression was negatively correlated with methylation level at cg01062029 and cg14145335 sites in TCGA-UCEC (*p* < 0.001). No significant correlation was observed for the cg03900646 site. (**B**). *ROR2* expression was not significantly different between adjacent stages in TCGA-UCEC. (**C**). *ROR2* expression was significantly lower in high grade compared to low grade EC in TCGA-UCEC (*p* < 0.001). (**D**). Methylation level of all the three CpG sites (cg01062029, cg03900646 and cg14145355) located in the *ROR2* promoter was significantly different between high and low grade EC (*p* < 0.001, *p* = 0.008, *p* < 0.001 respectively). (**E**). *ROR2* expression was significantly lower in high grade compared to low grade EC in GSE17025 (*p* = 0.004).

**Figure 3 cancers-13-00383-f003:**
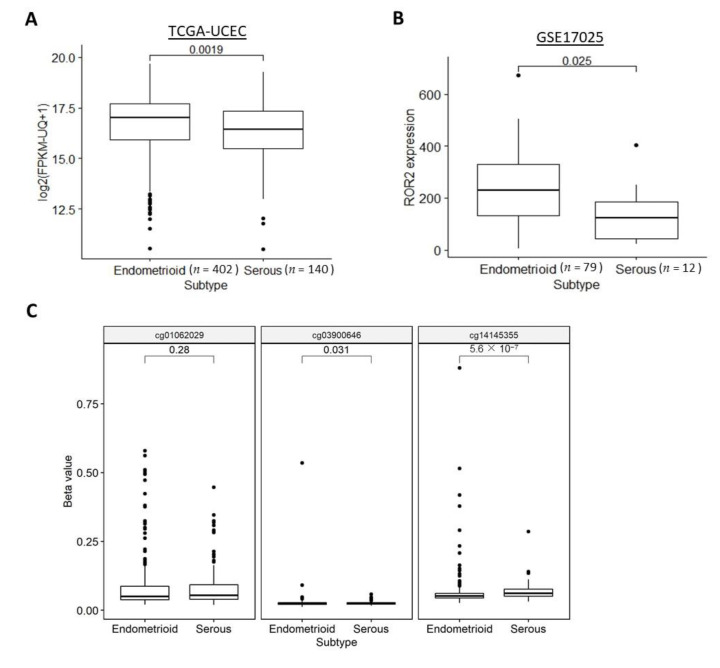
ROR2 mRNA expression level was downregulated in serous EC due to hypermethylation of ROR2 promoter region in the TCGA-UCEC and GSE17025 datasets. (**A**): *ROR2* expression level was significantly higher in endometrioid compared to serous EC in TCGA-UCEC (*p* = 0.002). (**B**): *ROR2* expression level was significantly higher in endometrioid compared to serous EC in GSE17025 (*p* = 0.025). (**C**): Methylation level of both cg03900646 and cg14145355 in the *ROR2* promoter was significantly different between serous and endometrioid EC (*p* = 0.031, *p* < 0.001 respectively).

**Figure 4 cancers-13-00383-f004:**
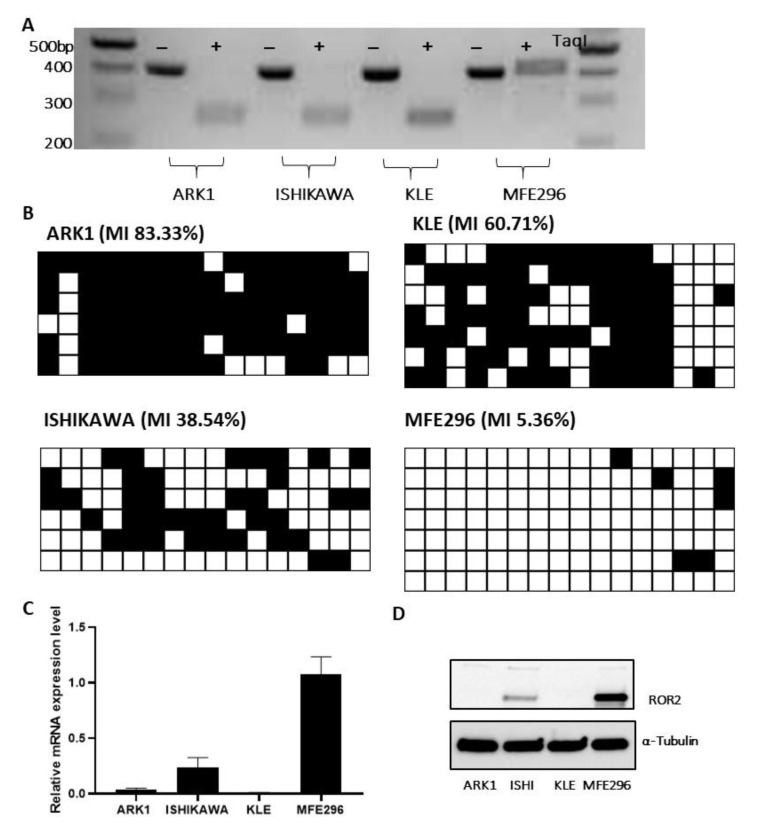
ROR2 expression was suppressed in the high grade and serous EC cell lines via hypermethylation within the *ROR2* promoter region. (**A**). COBRA assay of 4 EC cell lines (ARK-1, ISHIKAWA, KLE and MFE296) showing PCR amplicons (395 bp) with and without TaqI digestion. Methylation is detectable as digestion of the amplicon at 295 bp and 100 bp fragments. Samples without methylation exhibit no digestion. (**B**). Bisulphite sequencing of the four EC cell lines. Black squares represent methylated CpG dinucleotides. White squares represent unmethylated CpG dinucleotides. *ROR2* hypermethylation is observed in ARK-1 and KLE. MI = Methylation Index. (**C**). ROR2 mRNA expression in EC cell lines measured by qRTPCR and normalised to 3 housekeeping genes (*n* = 3). Error bar represents standard deviation. (**D**). ROR2 protein expression in EC cell lines analysed by Western blot. Normalised intensity of bands and whole blots were provided in Appendix A.

**Figure 5 cancers-13-00383-f005:**
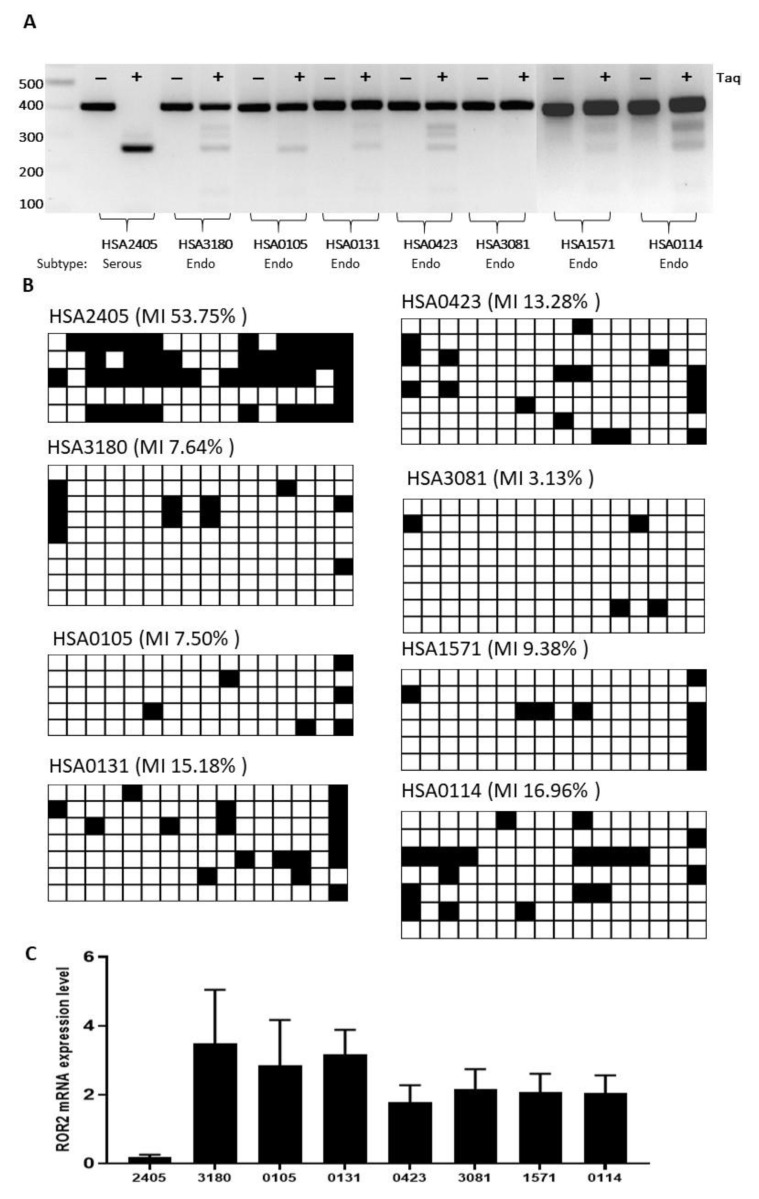
*ROR2* expression was correlated with promoter methylation index (MI) in EC patient samples. (**A**). COBRA assay of 8 patient samples showing PCR amplicons (395 bp) with and without TaqI digestion. HSA2405 sample showed complete digestion of the amplicon, which indicated existence of methylation. (**B**). Bisulphite genomic sequencing results of 8 EC patient samples. HSA2405 showed higher methylation level than other cases. (**C**). ROR2 mRNA expression in 8 EC patient samples as measured by qRTPCR and normalised to 3 housekeeping genes (*n* = 3). The serous EC patient HSA2405 which showed digestion in the COBRA assay and high methylation level showed low ROR2 expression by qRTPCR.

**Figure 6 cancers-13-00383-f006:**
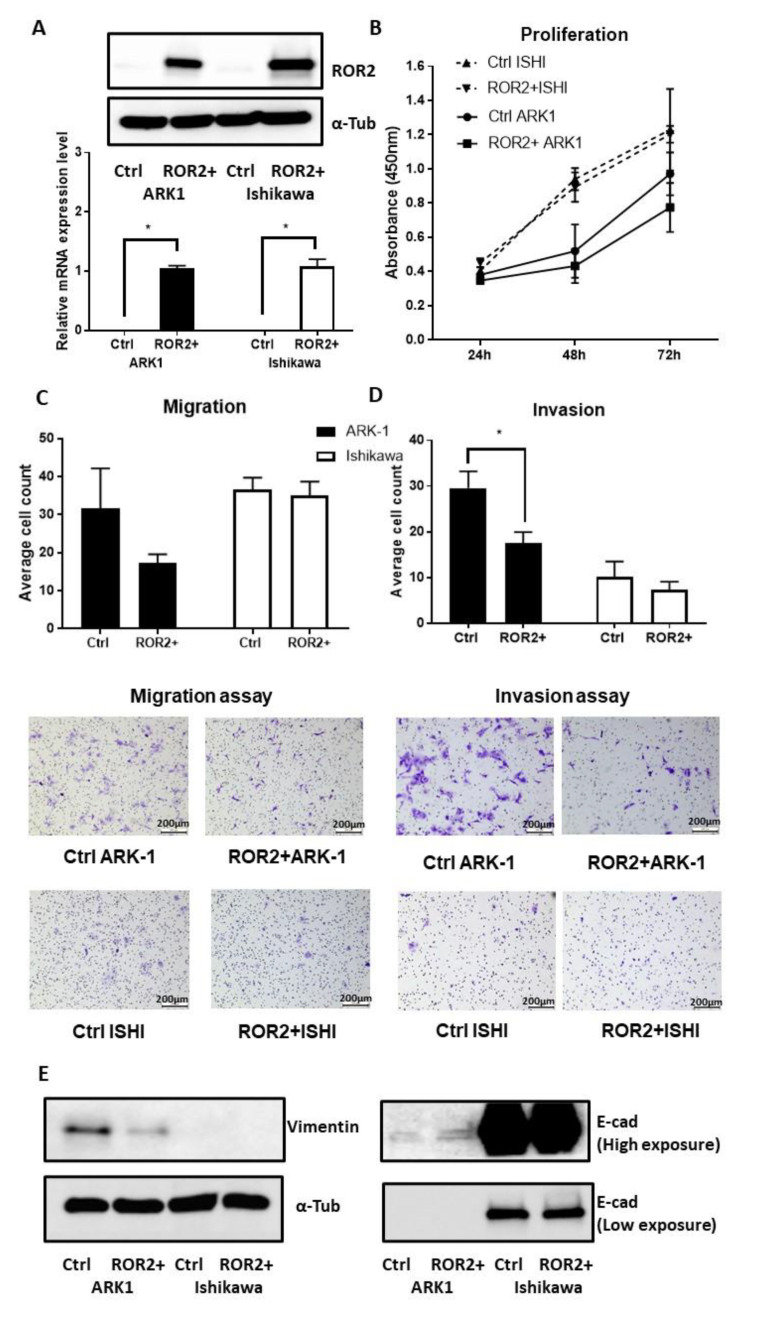
ROR2 overexpression significantly decreased the invasion ability of ARK-1 high grade serous EC cell. (**A**). ROR2 was significantly increased at both the transcriptional and translational levels following ROR2 plasmid transfection indicated by qRTPCR and Western blot. (**B**). ROR2 overexpression moderately decreased cell proliferation of ARK-1 48 h and 72 h after transfection (*p* = 0.470 and 0.150 respectively, *n* = 3). No significant effect was observed in Ishikawa following trans Figure 2. plasmid (*n* = 3) as well as representative images of transwell migration assay at 10× magnification. (**C**). Average invaded cell count between cells transfected with control plasmid and ROR2 plasmid (*n* = 3) as well as representative images of transwell migration assay at 10× magnification. (**D**). Average invaded cell count between cells transfected with control plasmid and ROR2 plasmid (*n* = 3) as well as representative images of transwell invasion assay at 10× magnification. (**E**). Western blot showing restoration of ROR2 in ARK-1 decreased Vimentin and increased E-cadherin level. Normalised intensity of bands and whole blots were provided in Appendix A. For all panels *n* = 3, error bars represent standard deviation of the mean, * Significant at *p* < 0.05 level.

**Table 1 cancers-13-00383-t001:** Summary of clinicopathological parameters collected from the HSA Biobank patients.

Patient ID	Histo Type	Stage	Grade
HSA2405	Serous	III	3
HSA3180	Endometrioid	IIIA	1
HSA0105	Endometrioid	IIIA	2
HSA0131	MMMT *	IB	3
HSA0423	Endometrioid	IV	1
HSA3081	Endometrioid	IA	3
HSA1571	Endometrioid	IA	3
HSA0114	Endometrioid	IA	2

* MMMT—Malignant Mixed Müllerian Tumour.

## Data Availability

Publicly available datasets were analysed in this study. This data can be found here: https://portal.gdc.cancer.gov/ and https://www.ncbi.nlm.nih.gov/geo/.

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
