# Peer review of "ROR2 Is Epigenetically Regulated in Endometrial Cancer"

_cancers, 2021, doi:10.3390/cancers13030383_

Round 1

Reviewer 1 Report

This is a well written paper that contains a thorough evaluation of ROR2 in endometrial cancer.  Ttthe authors have done a remarkable job of leaving no stone unturned. 

I would like a little ore discussion on the clinical significance roflcopters this finding.  How would they use this as a target and under what circumstances.

Would a demethtylating agent be useful interns of planning a trial or would this be just a maker?  

Please hypothesize as to what you think this is doing.

Author Response

  • Point 1: I would like a little more discussion on the clinical significance roflcopters this finding. How would they use this as a target and under what circumstances?
  • Response 1: The clinical significance of our study can be composed of two major aspects: Type II endometrial cancer screening and the identification of a novel therapeutic target. For the screening, we have added the following content in the discussion (Page 12, Line 304-308):

“A recent study demonstrated the potential application of DNA methylation profiles of certain biomarkers for EC screening of cervical scrapings [43]. However, the biomarkers identified in the study were based on Type I EC, which left the application in high grade or other Type II subtypes unclear. Our study proposes a promising biomarker for methylomics analysis in high grade and serous EC, which warrants future exploration.”

In addition, this study was designed following our previous findings in endometrial cancer which highlighted the effect of overexpression ROR2 in combination with inhibiting its sister receptor ROR1. There are currently several ROR1 targeting therapeutic options available while none for ROR2 overexpression. Therefore, we aimed to investigate the mechanism associated with the aberrant expression of ROR2 in endometrial cancer and thus find the potential way to restore the expression of ROR2. ROR1 targeting therapy in combination with ROR2 restoring therapy holds great potential in treating the high grade serous endometrial cancer patients who are usually diagnosed early but relapse. In response to the reviewer’s comment, we have included more discussion on the clinical significance in our discussion (Page 12, Line 311-316) as below:

Our previous in-vitro study in EC showed ROR2 overexpression in combination with silencing its sister receptor ROR1 inhibited the metastatic potential of KLE endometrial cancer cells more than either of the modifications alone [48].  With several ROR1 targeting therapies currently in development and in phase I clinical trials for other cancers, combination therapy may hold potential in patients with HGSEC, who currently have limited therapeutic options.

  • Point 2: Would a demethylating agent be useful interns of planning a trial or would this be just a maker? Please hypothesize as to what you think this is doing.
  • Response 2: Demethylation agents have been applied in clinical trials and showed promising outcomes in several other tumour types, but is limited in specificity. The global demethylation effect should be further investigated in endometrial cancer in the future. In addition, Type I and II should be trialled individually as they might show distinct responses to the agents. We have included more discussion on the future applications and concerns of the demethylating agents in the discussion (Page 12, Line 316-327) as below:

However, the functional outcome associated with the global demethylation effect remains unclear. Despite global hypomethylation of EC compared to normal endometrium, promoter hypermethylation of several genes such as MLH1 (DNA mismatch repair gene) [49], PTEN (Phosphatase and ten-sin homolog) [50] and APC (which regulates β-catenin Wnt signalling pathway) have been observed frequently in certain types of EC, especially Type I. Compared to Type I, Type II EC showed distinct DNA methylation profiles, including a different methylation pattern of DNMT (DNA methyltransferase gene) [51]. Therefore, divergent responses to the demethylation agents between Type I and II EC patients could be expected. Previous in-vivo studies have demonstrated promising effects of the demethylating agent 2’-deoxy-5-azacytidine (DAC) in restoring MLH1 expression and reversing resistance to chemotherapies in ovarian cancer [52]. There may be potential for combination treatment with chemotherapy and demethylation agents to treat chemoresistant EC patients, but further research into this area will be required.”

Reviewer 2 Report

The manuscript entitled “ROR2 is epigenetically regulated in endometrial cancer” by Liu et al. proposed that epigenetic regulation of ROR2 is associated with the development of endometrial cancer. The data are interesting and may provide some novel information to understand the role of ROR2-mediated Wnt signaling in endometrial cancer. This reviewer has some minor comments.

Comments

1) The authors previously demonstrated that ROR2 overexpression and ROR1 knockdown inhibited the migration, but not invasion, of endometrial cancer cell line KLE (Sci Rep. 2020; 10:13906). ROR2 promoter region is also hypermethylated and ROR expression is significantly decreased in KLE as well as ARK1. Why overexpression of ROR2 did not repress the invasion of KLE?

2) In addition to the above the report, the authors showed that ROR1 and ROR2 play distinct roles in endometrial cancer (Gynecol Oncol. 2018;148:576). This background should be described in the manuscript.

3) What is the actual ligand of ROR2 or ROR1 in endometrial cancer? This should be discussed in the manuscript.

Author Response

  • Point 1: The authors previously demonstrated that ROR2 overexpression and ROR1 knockdown inhibited the migration, but not invasion, of endometrial cancer cell line KLE (Sci Rep. 2020; 10:13906). ROR2 promoter region is also hypermethylated and ROR expression is significantly decreased in KLE as well as ARK1. Why overexpression of ROR2 did not repress the invasion of KLE?
  • Response 1: Our previous study investigating both ROR1 and ROR2 showed the combination of ROR2 overexpression with ROR1 knockdown had the most effect on inhibiting metastatic features of KLE than either of the modifications alone. There are currently several ROR1 targeting therapeutic options available while there are currently none for ROR2 overexpression. Therefore, we aimed to investigate the mechanism associated with the aberrant expression of ROR2 in endometrial cancer and thus find the potential means to restore the expression of ROR2. In our previous study, ROR2 overexpression alone showed a trend towards inhibiting KLE migration and invasion but was not statistically significant. The difference between KLE and ARK1 in response to ROR2 overexpression could be due to the different histotypes and metastatic potentials. Specifically, KLE is a poorly differentiated endometrial cancer cell line while ARK1 is a high grade serous endometrial cancer cell line. Whether KLE belongs to Type I or II has not been determined but ARK1 is the typical Type II endometrial cancer cell line that has been widely used. Therefore, we included the well-established cell lines for both Type I (Ishikawa) and Type II (ARK1) in this study for functional analysis. In addition, despite that KLE also shows hypermethylated ROR2 promoter region and low ROR2 expression, it could have a distinct metastatic potential and molecular profile compared to ARK1. But as KLE is not a typical Type II cell line, it was not the focus of this study. To link this study with our previous study, we have added the following content in our discussion (Page 12, Line 311-316):

Our previous in-vitro study in EC showed ROR2 overexpression in combination with silencing its sister receptor ROR1 inhibited the metastatic potential of KLE endometrial cancer cells more than either of the modification alone [48]. With several ROR1 targeting therapies currently in development and in phase I clinical trials for other cancers, combination therapy may hold great potential in patients with HGSEC, who currently have limited therapeutic options.

  • Point 2: In addition to the above the report, the authors showed that ROR1 and ROR2 play distinct roles in endometrial cancer (Gynecol Oncol. 2018;148:576). This background should be described in the manuscript.
  • Response 2: We appreciate the valuable recommendation suggested by the reviewer. We have included more background information in the introduction (Page 2, Line 68-72) as below:

This pathway can be activated through Wnt5a ligand binding to the tyrosine kinase-like orphan receptors ROR1 and ROR2. Different to the oncogenic role of the two receptors in ovarian cancer where both receptors are overexpressed and associated with survival [16], our previous study in EC suggested distinct roles for ROR1 and ROR2, with high ROR1 and low ROR2 expression associated with shorter survival [17].

  • Point 3: What is the actual ligand of ROR2 or ROR1 in endometrial cancer? This should be discussed in the manuscript.
  • Response 3: The precise ligands for the ROR1 and ROR2 tyrosine kinase-like orphan receptors in endometrial cancer is not yet established. However, previous studies in chronic lymphocytic leukemia (CLL) where ROR1 plays an oncogenic role showed that the Wnt5a ligand binds to ROR1/ROR2 heterooligomers and activates non-canonical Wnt signalling (Yu et al., 2016). But as ROR1 was not the focus of this study, we did not include the study in our manuscript. In this study, we hypothesised that ROR2 may dominate in β-catenin dependent Wnt signalling pathway in endometrial cancer similar to that in colon cancer. A key study in colon cancer suggested that the β-catenin dependent Wnt signalling mediated by ROR2 hypermethylation did not necessarily require Wnt5a (Lara et al., 2010). We have added more discussion (Page 12, Line 296-301) as below:

The presumed ligand for ROR2 in endometrial cancer is Wnt5a. However, a previous study in colon cancer showed restoration of the tumour suppressor ROR2 impaired tumour growth through inhibition of β-catenin dependent Wnt signalling, in a Wnt5a independent manner [25]. Therefore, ROR2 may dominate in either of the two arms of downstream Wnt signalling depending on the tumour context. Further exploration into the role of ROR2 in either arm of the Wnt signalling as well as its ligands is required in EC.”

Reviewer 3 Report

This is an excellent paper,providing convincing statistical, clinical and experimental evidence that ROR2 is a tumor suppressor in EC and that it is epigenetically regulated.My only concern is that among the clinical EC samples there is only one serous EC on which the whole argumentation of this part is based. To strenghten their conclusions the author should provide data from two or three more samples of serous EC, which should be easily available. They have included one sample of a MMMT , which according to newer findings is derived from serous EC.The authors might comment on this.

Author Response

  • Point 1: My only concern is that among the clinical EC samples there is only one serous EC on which the whole argumentation of this part is based. To strenghten their conclusions the author should provide data from two or three more samples of serous EC, which should be easily available.
  • Response 1: For this study we had access to only a small cohort of endometrial cancer patients with sufficient fresh tissue available for both RNA and DNA analysis. The range of subtypes reflects the clinical prevalence of these tumours locally, and unfortunately due to their rare nature and smaller Australian population our cohort only included one high grade serous EC case. However, the combination of the larger public available cohort with a smaller defined patient cohort allowed us to increase the statistical power for analysis. We accept that including more serous EC samples would strengthen our conclusion, therefore we have highlighted the limitation and proposed future directions in the discussion (Page 11, Line 269-275) as below:

In combination with the large public cohorts and well-established EC cell line models, this cohort confirmed deregulation of ROR2 expression as well as hypermethylation of ROR2 in serous EC. This study is limited in the number of primary serous EC samples available, but does include variable tumour grades and subtypes. Future investigation of a larger cohort of primary patient samples including more serous EC samples is warranted to uncover the epigenetic role in progression of EC.

  • Point 2: They have included one sample of a MMMT, which according to newer findings is derived from serous EC. The authors might comment on this.
  • Response 2: We appreciate the valuable recommendation suggested by the reviewer. We have added additional discussion on the MMMT results in our discussion (Page 11, Line 275-283) as below:

In addition, our clinical cohort incorporated one sample of Malignant Mixed Müllerian Tumour (MMMT). The majority of MMMT cases share common molecular features with serous EC while a less proportion similar to endometrioid subtype [38]. The DNA methylation profiles of MMMT also grouped the subtype into three clusters, with one similar to endometrioid EC and the other two resembles serous EC [39]. In our clinical cohort, the methylation and expression of ROR2 of the MMMT sample is similar to that of the endometrioid EC samples. However, due to the heterogeneity of this subtype, it would be helpful to include more MMMT subtypes in future studies to uncover epigenetic regulation mechanisms.

Round 2

Reviewer 3 Report

The problem of MMMTs has been answered satisfactorily. However , I think, it should be possible to analyze two or three additional serous EC samples. They are not that rare.

Author Response

We appreciate the comments suggested by the reviewer. Unfortunately, it is not possible for us to include any additional serous endometrial cancer samples currently as there was only one high grade serous endometrial cancer patient recruited in this local cohort. We did highlight this limitation in our manuscript and pointed out the need for validation in a future larger cohort. In order to obtain sufficient DNA and RNA for our experiments from 2-3 new high grade serous cases we would need to seek access to collaborators biobanks which would take considerable time to account for new ethics approvals, biobank applications and experimental work. We have made the conclusion based on the available data of the current cohort, without overstating the results. More importantly, we are confident that adding more samples will not change the results of our current data which should be taken in combination with the large-scale public cohort analysis and well-established endometrial cancer cell line models used.

This manuscript is a resubmission of an earlier submission. The following is a list of the peer review reports and author responses from that submission.

Round 1

Reviewer 1 Report

The manuscript provided by Liu D et al. nicely shows that expression of ROR2 is epigenetically controlled in endometrial cancer. Additionally they show that low ROR2 expression is correlated with unvavourable outcome and reduced overall survival.

The manuscript is well written and experiments seem to be adequately designed and conducted.

A few points need to be addressed or discussed:

1.) For people not familiar with endometrial cancer, abbreviations like POLE and MSI etc. should be explained in the text.

2.) For people not familiar with endometrial cancer, like me, the classifications used in the manuscript are puzzling. The introduction gives 2 classifications, with “copy number low (endometrioid)” and “copy number high (serous)” seem to be of importance.

Q: What does copy number mean here? Whole Genome? Polyploidy? Certain genes or chromosomes?

On page 4 l 98ff you compare serous and endometrioid high and low grade tumours,

on page 5 l 119ff you write “high grade or serous cases compared to low grade or endometroid”

on page 9 l you again use both of the above versions for the respective data set.

Q: One of these definitions seems to be wrong. At least they seem not compatible to me. Could you explain this? Why don’t you compare the same in both sets?

3.) In the introduction l. 38ff you state a survival rate of 80% for endometrial cancers mainly through radical surgical intervention and that survival outcomes for high grade...and highly aggressive subtypes remain poor.

Q: Which patients do you think could profit from your results? The ones with highly aggressive and/or (?) high grade? Did you check the data-sets for “aggressive” vs. not or “metastasized vs. not metastasized”?

It would be helpful if this could be picked up more in detail in the discussion.

4.) In the discussion you also state that a part of your data are not valid for serous EC due to biased distribution of the samples.

If you are aware of this, why do you present the data here? In my opinion the manuscript should focus on statistically valid data.

Author Response

  • Point 1: For people not familiar with endometrial cancer, abbreviations like POLE and MSI etc. should be explained in the text.
  • Response 1: In response to the reviewer’s comment, we have expanded the abbreviation of MSI-hypermutated to “microsatellite instability hypermutated” (Line 41). In addition, POLE is a gene that encodes the catalytic subunit of DNA (Line 40). We have italicised the gene name to make it clear.
  • Point 2: For people not familiar with endometrial cancer, like me, the classifications used in the manuscript are puzzling. The introduction gives 2 classifications, with “copy number low (endometrioid)” and “copy number high (serous)” seem to be of importance. Q: What does copy number mean here? Whole Genome? Polyploidy? Certain genes or chromosomes?

On page 4 l 98ff you compare serous and endometrioid high and low grade tumours,

on page 5 l 119ff you write “high grade or serous cases compared to low grade or endometroid”

on page 9 l you again use both of the above versions for the respective data set.

Q: One of these definitions seems to be wrong. At least they seem not compatible to me. Could you explain this? Why don’t you compare the same in both sets?

  • Response 2: The copy number high group is characterised by a very high degree of somatic copy number alterations by whole genome sequencing. The four subgroups of endometrial cancer were identified by the Cancer Genome Atlas (TCGA) Network after integrating genomic, transcriptomic and proteomic characterisation of endometrial cancer tissue. This classification has not been widely used diagnostically. In this study, the histological subtype (mainly consisting of endometrioid and serous subtypes) and tumour grade (high and low grades) were used for stratification. To clarify this point, the introduction has now been updated to include the relationship between the TCGA classifications and histological subtypes (Line 42) as below:

“The copy-number high cluster included most of the serous and serous-like endometrioid subtypes while the other three clusters mainly consisted of the endometrioid subtypes.”

  • Point 3: In the introduction l. 38ff you state a survival rate of 80% for endometrial cancers mainly through radical surgical intervention and that survival outcomes for high grade...and highly aggressive subtypes remain poor.

Q: Which patients do you think could profit from your results? The ones with highly aggressive and/or (?) high grade? Did you check the data-sets for “aggressive” vs. not or “metastasized vs. not metastasized”?

It would be helpful if this could be picked up more in detail in the discussion.

  • Response 3: We have extracted the progression data from the TCGA cohort and performed the progression free survival analysis of ROR2. No significant association between ROR2 and relapse was observed in the TCGA cohort (Supplementary Figure 1E, p=0.084). There is no aggressiveness data available from the public cohort, but the serous subtype or high tumour grade is commonly regarded as highly aggressive subgroup of EC. Therefore, we also performed an additional cox regression correlation analysis on both high grade and serous subgroups of the TCGA-UCEC cohort. The results showed methylation of CpG sites on ROR2 promoter, rather than ROR2 expression level, has a significant correlation with overall survival in both high grade and serous EC subgroups. A significantly negative correlation was also observed between ROR2 expression level and methylation of CpG sites of the ROR2 promoter in both high grade and serous EC (Supplementary Figure 1C,D). We have added a supplementary figure and corresponding description in section 2.1 (Line95-101) as below:

“In the high-grade subgroup of the cohort, methylation level of two CpG sites cg14145355 and cg03900646, rather than ROR2 expression level, were significantly associated with overall survival after filtering against other variables including age, BMI, stage. (Supplementary Figure 1A, p=0.013and 0.014 respectively). In contrast, one CpG site cg01062029 was significantly associated with overall survival in serous EC patients of the cohort (Supplementary Figure 1 B, p=0.033). No significant correlation was observed for ROR2 expression level with progression free survival (Supplementary Figure 1 E, p=0.084).” 

  • Point 4: In the discussion you also state that a part of your data are not valid for serous EC due to biased distribution of the samples.

If you are aware of this, why do you present the data here? In my opinion the manuscript should focus on statistically valid data.

  • Response 4: Our bioinformatics analyses were based on the most validated public cohorts currently available for EC. The reason of the limitation of the distribution is because the majority of serous EC is high grade.

Reviewer 2 Report

The stated aim of this study was to investigate the role of ROR2 in endometrial cancer (EC) and the mechanism associated with its altered expression.  Rather than studying the role of ROR2 in EC, the authors actually only provide data regarding expression of ROR2 in EC.  No functional analyses of the role of ROR2 in EC were provided, not even Western blot analysis of Wnt pathway members.  In this reviewer’s opinion, the authors lay the foundation for an interesting study of ROR2 expression in EC but do not provide enough experimental evidence to back their overall conclusions and tend overstate their results throughout the manuscript.  Most notably, they fail to provide a convincing correlation between ROR2 promoter methylation and mRNA expression in high grade and serous EC and never functionally prove promoter methylation leads to decreased expression of ROR2. Specific critiques are:

  • Title: Serous endometrial cancers are by definition high grade tumors so is unclear whether mRNA expression is actually correlated with serous histology AND high grade or if this is simply stating the same result two different ways. Based on this, I suggest that the title be revised to: ROR2 mRNA expression is downregulated in high grade endometrial cancer, potentially through promoter methylation
  • Abstract Lines 22-23: the authors refer to the KLE cell line as a serous endometrial cancer cell line. The original publication for this cell line does not define the originating tumor as serous histology (Richardson GS et al.  KLE: a cell line with defective estrogen receptor derived from undifferentiated endometrial cancer.  Gynecologic Oncology 1984; 17: 213-230).
  • Figure 4 Panel B- Authors should add size markers to figure and intended protein size indicated.  The figure should also not be cropped so tightly and it should be indicated if authors expected multiple bands.
  • Figure 4 Panels C and D- The authors should state if there a significant correlation between the COBRA assay results and RNA expression of ROR2.
  • The authors should add clarity as to why they choose 3 most likely endometrioid cell lines, 7 endometrioid primary tumors and one MMMT primary tumor to investigate the extent of methylation across the ROR2 promoter rather than serous cell lines and primary tumors? Did the extent of methylation in these samples correlate with grade?  Although the KLE cell line is not recognized as a serous cell line, contrary to claims in this manuscript, there are a number of known serous endometrial cancer cell lines available and this manuscript would benefit greatly from the addition of serous cell lines and primary tumors.
  • Figure 6B: Is the “negative correlation” between mRNA expression and promoter methylation index (MI) driven by the single sample with MI of just over 40?
  • Supplemental Figure 1- Panel A shows that ROR2 expression was significantly lower in tumor compared to normal endometrium but panel B shows no difference in methylation. It is potentially misleading that this is not mentioned in lines 89-90 of the main text.
  • Supplemental Figure 3- Regarding panel A, what was the expected protein size of ROR2 and were all bands used in the quantification shown in panel B? What do the 4 bands on the left side of panel C represent?
  • Is ROR2 expression prognostic for survival in serous or high-grade EC?
  • Discussion lines 173-178: This paragraph begins with “In this study,… and ands with, “On top of this we investigated the effect of manipulation of ROR2 expression levels on the metastatic potential of endometrial cancer cell lines.” Where is this metastasis data presented in the current manuscript?
  • Discussion lines 191-192, line 195: as previously stated, the original manuscript describing the derivation of the KLE cell line does not define these cells as serous.
  • Methods line 256- is DMEM/F13 a typo; did the authors mean DMEM/F12?
  • Conclusion (lines 312-314): Is this the first study to investigate the role of ROR2 in EC as stated? The aim of a previous publication by these authors (reference #24) was “to investigate the role of ROR1 and ROR2 in EC in a larger Australian population-based EC cohort…”.  The authors also state that ROR2 may be a diagnostic or therapeutic candidate for EC.  Figure 1 Panel E shows mRNA expression of ROR2 does not significantly correlate with OS in EC, which is in line with previous publications by this group; likewise, the human protein atlas shows that ROR2 protein expression is not prognostic in EC.  The authors should clarify what data back their statement that ROR2 may be a diagnostic or therapeutic candidate for EC. 

Author Response

  • General comment: The stated aim of this study was to investigate the role of ROR2 in endometrial cancer (EC) and the mechanism associated with its altered expression.  Rather than studying the role of ROR2 in EC, the authors actually only provide data regarding expression of ROR2 in EC.  No functional analyses of the role of ROR2 in EC were provided, not even Western blot analysis of Wnt pathway members.  In this reviewer’s opinion, the authors lay the foundation for an interesting study of ROR2 expression in EC but do not provide enough experimental evidence to back their overall conclusions and tend overstate their results throughout the manuscript.  Most notably, they fail to provide a convincing correlation between ROR2 promoter methylation and mRNA expression in high grade and serous EC and never functionally prove promoter methylation leads to decreased expression of ROR2.
  • General response: Many of these points have been addressed individually below. However, we would like to note that we have now performed additional analysis on the high grade and serous subgroup of TCGA cohort (Supplementary Figure 1), which supports the negative correlation between ROR2 promoter methylation and mRNA expression in high grade and serous EC patients. In addition, we tempered our statements regarding serous subtype of EC throughout the manuscript.
  • Point 1: Title: Serous endometrial cancers are by definition high grade tumors so is unclear whether mRNA expression is actually correlated with serous histology AND high grade or if this is simply stating the same result two different ways. Based on this, I suggest that the title be revised to: ROR2 mRNA expression is downregulated in high grade endometrial cancer, potentially through promoter methylation
  • Response 1: We appreciate this point of view, and have removed reference to “serous’ from our title. Because our manuscript includes analysis of CpG sites as well as mRNA we do not believe the suggested title is reflective of the manuscript. Therefore, we propose a new title based on this feedback as “ROR2 is epigenetically inactivated in high grade endometrial cancer.”
  • Point 2: Abstract Lines 22-23: the authors refer to the KLE cell line as a serous endometrial cancer cell line. The original publication for this cell line does not define the originating tumor as serous histology (Richardson GS et al. KLE: a cell line with defective estrogen receptor derived from undifferentiated endometrial cancer.  Gynecologic Oncology 1984; 17: 213-230).
  • Response 2: We have changed “serous” to “high grade” when describing the KLE cell line throughout the paper.
  • Point 3: Figure 4 Panel B- Authors should add size markers to figure and intended protein size indicated. The figure should also not be cropped so tightly and it should be indicated if authors expected multiple bands.
  • Response 3: Panel B has been adjusted with addition of molecular size information and blot image with expanded height. The figure legend has also been updated with additional information (Line 175-177) as: “Molecular weight of each band was calculated by the ImageQuant TL software based on the marker loaded on the blot. The expected size is 104.76 and 50kDa for ROR2 and α-Tubulin respectively.”
  • Point 4: Figure 4 Panels C and D- The authors should state if there a significant correlation between the COBRA assay results and RNA expression of ROR2.
  • Response 4: We have performed the correlation analysis of ROR2 expression and COBRA assay and added in the manuscript in Line 168-169 as “ROR2 expression is significantly correlated with the COBRA digestion status (p=0.011).” and updated corresponding method in section 4.10. Statistical analysis (Line 350) as “Correlation between expression and COBRA assay results was analysed through unpaired t-test.
  • Point 5: The authors should add clarity as to why they choose 3 most likely endometrioid cell lines, 7 endometrioid primary tumors and one MMMT primary tumor to investigate the extent of methylation across the ROR2 promoter rather than serous cell lines and primary tumors? Did the extent of methylation in these samples correlate with grade? Although the KLE cell line is not recognized as a serous cell line, contrary to claims in this manuscript, there are a number of known serous endometrial cancer cell lines available and this manuscript would benefit greatly from the addition of serous cell lines and primary tumors.
  • Response 5: For this study we had access to only a small cohort of endometrial cancer patients with sufficient fresh tissue available for both RNA and DNA analysis. The range of subtypes reflects the clinical prevalence of these tumours locally, and unfortunately due to their rare nature our cohort did not include high grade serous subtypes. Despite this limitation the cohort does include tumour grades 1 to 3, and despite its small size does show that hypermethylation of the ROR2 promoter is associated with ROR2 suppression in EC patient samples.

In addition, the combination of the larger public available cohort with a smaller defined patient cohort allowed us to increase the statistical power for analysis. We accept that in our original manuscript we have overstressed the implications of our results for high grade serous endometrial cancer, and have adjusted our text throughout and in response to the reviewer’s other comments.

  • Point 6: Figure 6B: Is the “negative correlation” between mRNA expression and promoter methylation index (MI) driven by the single sample with MI of just over 40?
  • Response 6: Yes, the one sample which showed a high MI level could have an effect on the correlation analysis. We checked this by removing the sample, and the negative correlation remained (R=-0.138).
  • Point 7: Supplemental Figure 1- Panel A shows that ROR2 expression was significantly lower in tumor compared to normal endometrium but panel B shows no difference in methylation. It is potentially misleading that this is not mentioned in lines 89-90 of the main text.
  • Response 7: Description of Supplementary Figure 2 panel B was added in Line 112-114 as “However, no significant difference was observed between tumour and normal tissue in the methylation level of any of the three CG sites within ROR2 promoter (Supplementary Figure 2 B).” following that of panel A.
  • Point 8: Supplemental Figure 3- Regarding panel A, what was the expected protein size of ROR2 and were all bands used in the quantification shown in panel B? What do the 4 bands on the left side of panel C represent?
  • Response 8: The expected protein size of ROR2 is 104.76kDa. The relative ROR2 expression level was calculated based on the intensity of ROR2 (band at 105kDa) normalised against that of α-Tubulin. The four bands on the left side of panel C represent the same four samples loaded for another protein analysis which was not included in this study. We have included the entire membrane as requested by the journal. Molecular size of ROR2 bands and labels of the panel C samples have been updated.
  • Point 9: Is ROR2 expression prognostic for survival in serous or high-grade EC?
  • Response 9: The overall survival analysis of ROR2 in serous and high-grade subgroups of TCGA has been performed and included in Supplementary Figure 1. No significant association was observed for ROR2 expression level but methylation of CG probes showed significant correlation with overall survival in both high grade or serous subgroups. The corresponding description has also been added in section 2.1 (Line 95-100) as below:

“In the high-grade subgroup of the cohort, methylation level of two CpG sites cg14145355 and cg03900646, rather than ROR2 expression level, were significantly associated with overall survival after filtering against other variables including age, BMI, stage. (Supplementary Figure 1A, p=0.013and 0.014 respectively). In contrast, one CpG site cg01062029 was significantly associated with overall survival in serous EC patients of the cohort (Supplementary Figure 1 B, p=0.033).”

  • Point 10: Discussion lines 173-178: This paragraph begins with “In this study,… and ands with, “On top of this we investigated the effect of manipulation of ROR2 expression levels on the metastatic potential of endometrial cancer cell lines.” Where is this metastasis data presented in the current manuscript?
  • Response 10: This is an error. The sentence “On top of this we investigated the effect of manipulation of ROR2 expression levels on the metastatic potential of endometrial cancer cell lines.” has now been removed.
  • Point 11: Discussion lines 191-192, line 195: as previously stated, the original manuscript describing the derivation of the KLE cell line does not define these cells as serous.
  • Response 11: We have removed the “serous” for classification of the KLE cell line throughout the paper.
  • Point 12: Methods line 256- is DMEM/F13 a typo; did the authors mean DMEM/F12?
  • Response 12: The typo has been corrected.
  • Point 13: Conclusion (lines 312-314): Is this the first study to investigate the role of ROR2 in EC as stated? The aim of a previous publication by these authors (reference #24) was “to investigate the role of ROR1 and ROR2 in EC in a larger Australian population-based EC cohort…”. The authors also state that ROR2 may be a diagnostic or therapeutic candidate for EC.  Figure 1 Panel E shows mRNA expression of ROR2 does not significantly correlate with OS in EC, which is in line with previous publications by this group; likewise, the human protein atlas shows that ROR2 protein expression is not prognostic in EC.  The authors should clarify what data back their statement that ROR2 may be a diagnostic or therapeutic candidate for EC.
  • Response 13: Our study is the first to both investigate the role of ROR2 and its potential epigenetic regulation mechanisms in EC. The previous study by our group mainly focused on the role of ROR2 in EC but did not include regulation mechanism associated with its expression. Our multivariable overall survival analysis on TCGA cohort showed the methylation level of the two CpG sites located on ROR2 promoter was significantly associated with overall survival. Therefore, the methylation level, rather than the expression level of ROR2 could be a potential prognostic marker for overall survival of EC patients. Besides, TCGA and GEO datasets consistently showed ROR2 was significantly downregulated in high grade and serous subtype compared to low grade and endometrioid subtype (Figure 2A,C and Figure 3C,D). In addition, ROR2 expression was negatively correlated with promoter methylation level significantly in the complete cohort (Figure 2F) as well as high-grade and serous subgroups of TCGA cohort (Supplementary Figure 1C,D) and our patient cohorts (Figure 6B). Therefore, we conclude that ROR2 appears to play a tumour suppressor role, with its expression potentially regulated through promoter methylation. Therefore, demethylation agents could be potentially effective on treating patients with high grade serous EC.

Reviewer 3 Report

The authors present a clear analysis of the availble TCGA database with respect to ROR2 expression and compare these findings against their patient cohort. Their use of both the COBRA method to detect methylation anf RT-PCR to validate expression reaffirms their claim that ROR2 expression can be methylated in endometrial cancer. A clear and concise paper.

Author Response

We would like to acknowledge the time and effort for the reviewers spending on our manuscript and we honestly appreciate the positive feedback of the reviewer on our manuscript. 

Round 2

Reviewer 2 Report

Response to Author’s Responses:

  • General Response 1: “Many of these points have been addressed below” is not a sufficient response in my opinion.
  • Response 2: It is unclear why the authors initially designated the KLE cell line as serous but then just responded to my critique by removing this designation throughout the manuscript, which has led to a general feeling of unease with this reviewer regarding the entire manuscript. Furthermore, in starting to re-read the manuscript, despite the fact that I recommended rejection initially, I came across lines 44-45 of the introduction, which states: “Although the TCGA classification hold great potential, 40% of all EC cases could not be assigned to a specific molecular subgroup” with a reference for a review article with a sentence describing the No Specific Molecular Subtype, a designation assigned by the TransPORTEC initiative in manuscripts published following TCGA’s molecular classification of endometrial cancers where they attempt to pragmatically recapitulate TCGA’s molecular classifications.  I truly don’t believe the main author of this manuscript has a firm enough grasp of the genetics and biology of endometrial cancer to write an accurate introduction and I worry about the validity of the entire manuscript, particularly given the understandable confusion expressed by the reviewer not familiar with endometrial cancer.  The third reviewer didn’t appear to exert sufficient effort to review this manuscript.  I stopped trying to re-read the manuscript in it’s entirety and focused instead on the responses to my initial reviews, the majority of which were not sufficiently addressed as pointed out here.
  • Response 3: Figure 4 Panel B- in the revised still too tight cropping of the Western results, the authors now cropped what I imagine are non-specific bands previously visible above the band of interest. Authors indicate that the revised version has an expanded height which is not actually true, it’s an adjusted lower view.
  • Response 4: Figure 4 Panels C and D- The authors did not provide sufficient methodology to describe how they determined that the COBRA assay results correlated to mRNA expression.  Simply looking at the figure as a whole, starting with Panels A and B, they are trying to convince readers that the high protein expression in MFE-296 (Panel B) is a result of the methylation status shown in panel A (while ignoring nearly equivalent results for Ishikawa and KLE in panel A but protein expression in Ishikawa in Panel B but none in KLE).  Given the rational of the MFE-296 result, one would expect HSA03081 in panel C to have high protein expression of ROR2 in panel D but now Panel D shows mRNA expression which still is not high in HSA03081 compared to the other tumors.
  • Response 5: the authors did not address my comment regarding the availability of high grade cell lines (including serous). Also, using 8 endometrial cancers, only 3 of which are high grade, to use for functional validation of manuscript entitled “ROR2 is epigenetically inactivated in high grade endometrial cancer” is not sufficient, particularly given the fact that the authors never acknowledge this shortcoming in the manuscript.
  • Response 6: The author’s response is wildly insufficient. What is the p-value of the “negative correlation that remained R=-0.318”?  I don’t believe any rational scientist would look at that figure and believe there is a negative correlation present if the outlier is removed.
  • Response 7: response is sufficient, thank you.
  • Response 8: The results presented in this supplemental figure along with the original Figure 4 Panel B both show what this reviewer can only assume are multiple non-specific bands, leaving one to question the reliability of the antibody used.
  • Response 9: the authors fail to show a consistent correlation between any ROR2 protein expression, mRNA expression or methylation and survival, or a correlation between methylation and protein/mRNA expression. Throughout the manuscript they simply highlight which variables turned out to be (often time slightly) significantly correlated in whichever cohort they are highlighting at the moment, leading to a confusing and not convincing story.
  • Response 10: thank you for this response- This was another interesting unexplained oversight, leaving this reviewer to now wonder if ROR2 expression affected metastatic potential of the cell lines.
  • Response 13: This reviewer is not convinced by the data presented that ROR2 represents a valid potential diagnostic or therapeutic candidate for endometrial cancer. Furthermore, in the revised manuscript the authors write “To sum up, ROR2 plays a tumor suppressor role in EC”.  This has not been established.